# Comprehensive Analysis of the Characteristics and Differences in Adult and Newborn Brown Adipose Tissue (BAT): Newborn BAT Is a More Active/Dynamic BAT

**DOI:** 10.3390/cells9010201

**Published:** 2020-01-14

**Authors:** Junyu Liu, Chuanhai Zhang, Boyang Zhang, Yao Sheng, Wentao Xu, Yunbo Luo, Xiaoyun He, Kunlun Huang

**Affiliations:** 1Beijing Advanced Innovation Center for Food Nutrition and Human Health, College of Food Science and Nutritional Engineering, China Agricultural University, Beijing 100083, China; liujunyu0306@163.com (J.L.); zhangchuanhai.68@163.com (C.Z.); boyang0214@126.com (B.Z.); shengyao314@163.com (Y.S.); xuwentao@cau.edu.cn (W.X.); lyb@cau.edu.cn (Y.L.); 2Key Laboratory of Safety Assessment of Genetically Modifed Organism (Food Safety), Ministry of Agriculture, Beijing 100083, China

**Keywords:** neonatal, brown adipose tissue, activation, post-delivery development

## Abstract

Brown adipose tissue (BAT) plays an essential role in maintaining body temperature and in treating obesity and diabetes. The adult BAT (aBAT) and neonatal BAT (neBAT) vary greatly in capacity, but the characteristics and differences between them on the molecular level, as well as the related features of BAT as it develops post-delivery, have not yet been fully determined. In this study, we examined the morphological features of aBAT and neBAT of mice by using hematoxylin-eosin (H&E) staining, transmission electron microscopy (TEM), and scanning electron microscopy (SEM). We found that neBAT contains a smaller number and size of lipid droplets, as well as more abundant mitochondria, compared with aBAT. The dynamic morphological changes revealed that the number and size of lipid droplets increase, but the number of mitochondria gradually decrease during the post-delivery development, which consisted of some differences in RNA or protein expression levels, such as gradually decreased uncoupling protein 1 (*UCP1*) expression levels and mitochondrial genes, such as mitochondrial transcription factor A (*Tfam*). The adipocyte differentiation-related genes, such as transcription factor CCAAT enhancer-binding protein β (*CEBPβ*), were also continuously upregulated. Additionally, the different features of aBAT and neBAT were analyzed from the global transcription (RNA-Seq) level, which included messenger RNA (mRNA), microRNA, long non-coding RNA (lncRNA), circRNA, and DNA methylation, as well as proteins (proteomics). Differentially methylated region (DMR) analysis identified 383 hyper- and 503 hypo-methylated genes, as well as 1221 new circRNA in ne-BAT and 1991 new circRNA in a-BAT, with significantly higher expression of circRNA in aBAT compared with neBAT. Gene ontology (GO) enrichment analysis and Kyoto Encyclopedia of Genes and Genomes (KEGG) pathway analysis indicated that mitochondrial activity, protein synthesis, and cell life activity levels were higher in neBAT, and pathways related to ribosomes, spliceosomes, and metabolism were significantly activated in neBAT compared to aBAT. Collectively, this study describes the dynamic changes occurring throughout post-delivery development from the morphological, molecular and omics perspectives. Our study provides information that may be utilized in improving BAT functional activity through gene regulation and/or epigenetic regulation.

## 1. Introduction

Brown adipose tissues (BATs), which are abundant in mammals and neonates with abundant mitochondria and high thermoregulatory demands [1], have been recognized as a great potential star target organ for the treatment of obesity and related metabolic diseases. Previously, it was widely believed that BATs are present only in small mammals and human infants. Currently, there is growing evidence demonstrating that functional BATs are present in adult humans and its activity is negatively correlated with body mass index [2,3,4]. BATs play a role in maintaining body temperature against a cold environment by activating mitochondrial uncoupling protein 1 (*UCP1*) and are crucial to mammalian fetal development. The amount and distribution range of BATs gradually decrease with age. The recruitment and activation of BATs can be modulated by external stimuli, including environmental factors and endogenous factors [5]. The activity of BATs is higher during fetal development than in adulthood. However, no comparative analysis of BATs in the fetal and adult stages has been conducted to date.

Numerous studies have shown that the development of BATs is regulated by transcriptional control [6,7,8,9,10,11,12]. Peroxisome-proliferator-activated receptor γ(*PPARγ*) [13] and CCAAT/enhancer-binding proteins, including C/EBPα/β/δ(C/EBPs) [14], have been established as essential in the formation of both mature brown and white adipocytes [12]. While *PPARγ* co-activator 1α (*PGC1α)*, a nuclear co-activator [15], and PR-domain-containing 16 (PRDM16), which physically interacts with *C/EBPβ* [16,17], specifically regulate brown adipogenic programming, PRDM16 determines the switch between BAT and skeletal muscles [18]. The developmental ancestry of BAT has been extensively studied and collectively indicates that BAT has a closer relationship to skeletal muscles rather than white adipose tissues [8,18,19,20,21,22]. Furthermore, *Myf5*+ and *EN1*+ progenitor cells generate classic brown adipocytes as well as myoblasts [18,23,24]. Several studies suggested that *Myf5* is a location marker rather than a specific cell lineage maker [25,26,27]. *EN1* is not only a progenitor marker but also an activator of brown adipogenesis [24]. Also, there are several morphogenic signals, including bone morphogenetic protein (BMP), fibroblast growth factor (FGF), Wnt, and Hedgehog signaling pathways involved in the formation of brown adipocytes [28,29,30,31,32,33]. Moreover, several studies reported that members of the *TGFβ* (transforming growth factor β) superfamily hold distinct regulatory effects on brown adipogenesis [34,35]. In addition to the regulation of transcription levels, epigenetic regulation, including microRNA [36], lncRNA [37,38,39,40], and methylation [41,42], also play essential roles in the formation and activation of BAT [43]. Although circRNA has been rarely reported in the regulation of brown adipogenesis and functions, we believe it is also a very promising potential regulator.

In the current study, we examined the internal and surface morphology of aBAT and neBAT through transmission electron microscopy (TEM) and scanning electron microscopy (SEM), respectively, and checked the expression of brown adipogenic and thermogenic genes. We found a variety of significant differences between aBAT and neBAT at the morphological and molecular levels, which provided enough value to continue to explore their differences in protein, and epigenetic changes, including mRNA, microRNA, lncRNA, circRNA, and DNA methylation, were analyzed by proteomics, whole transcriptomics, and reduced representation bisulfite sequencing (RRBS), respectively.

In summary, we comprehensively analyzed the characteristics and differences in adult and newborn brown adipose tissues and found numerous differences and interesting findings, which may provide new insights into the treatment of metabolic diseases, such as reprogramming the low-active adult BAT into the more active newborn-like BAT. Importantly, our study analyzed the complete transcription and proteome of two kinds of BATs, which provides important information for understanding the feature of BATs and/or for developing a new method for improving BATs’ functional activity through gene regulation and/or epigenetic regulation.

## 2. Materials and Methods

### 2.1. Animal Care and In Vivo Experiment Procedures

C57BL/6 breeding pair mice (8-week-old) were obtained from Beijing Vital River Laboratory Animal Technology. For the experiments, the 0-, 2-, 4-, 6-, and 8-week-old male C57BL/6J mice were obtained from the breeding pair mice, and all post-weaning mice were housed (≤5 animals/cage) at 22 ± 2 °C and 55% ± 10% humidity with a 12-h light-dark cycle in an office of a Laboratory Animal Welfare-certified animal facility. Water and food were provided ad libitum. In this study, the following groups were used: BAT in 0-week-old (1–2 days after delivery) male mice, which refers to the newborn and kept-suckling group, named neBAT; BAT in 8-week-old (after delivery) male mice, which refers to the adult group, designated as aBAT; and BAT in mice with an embryonic stage of about 19 days, which refers to the embryonic group, named ME-BAT. The number of mice in each group was 6. After sacrificing the mice, we collected the BAT samples from the interscapular region. All experimental procedures and use of animals were conducted according to the Guide for the Care and Use of Laboratory Animals published by the US National Institute of Health and approved by the Animal Ethics Committee of China Agricultural University, Beijing (the approval ID is KY1700014).

### 2.2. Hematoxylin and Eosin Staining

Tissues fixed with 4% paraformaldehyde were sliced in paraffin. Hematoxylin-eosin staining was used for the preparation of multiple sections. Slices were placed into hematoxylin solution and dyed for several minutes, and color separation in acid water and ammonia water occurred for several seconds, respectively. Slices were rinsed with running water for 1 h and then distilled water was added for a while. Slices were dehydrated in 70% and 90% alcohol for 10 min, respectively, then dyed in eosin staining solution for 2 to 3 min. The stained section was dehydrated by pure alcohol and then penetrated by xylene. Finally, the transparent section was dropped with gum and sealed with a cover glass.

### 2.3. Transmission Electron Microscopy

BAT sections were fixed with 2% (*vol*/*vol*) glutaraldehyde in 100 mM phosphate buffer (pH 7.4) for 12 h at 4 °C. Sections were then post-fixed with 1% osmium tetroxide, dehydrated in ascending gradations of ethanol, and embedded in fresh epoxy resin 618 [44]. Ultrathin sections (60–80 nm) were cut and stained with lead citrate before being examined on a Hitachi H-7500 transmission electron microscope.

### 2.4. Scanning Electron Microscopy

BAT sections were fixed with 2% (*vol*/*vol*) glutaraldehyde in 100 mM phosphate buffer (pH 7.2) for 12 h at 4 °C. Sections were then adhered to the double-sided adhesive, adhered to the copper table, and then coated with conductive adhesive (nano gold) before being examined on a Hitachi H-7500 scanning electron microscope [45]. The size of lipid droplets and mitochondria were presented in the form of a particle size distribution map, which was completed by software Image-pro plus and OriginPro 8.

### 2.5. Quantitative Real-Time Reverse-Transcription PCR

Total RNA was isolated using a total tissue RNA isolation kit (ET101-01, TransGen Biotech, Beijing, China). Equal amounts of total RNA were used to synthesize cDNA with the transScript One-Step gDNA Removal and cDNA Synthesis SuperMix kit (AT311-03, TransGen Biotech, Beijing). Quantitative real-time reverse-transcription PCR (qRT–PCR) was performed in triplicate using SYBR Green, 96-well plates, and the Real-Time PCR System (Bio-Rad, Hercules, CA, USA). Each well was loaded with a total of 20 μL containing 2 μL of cDNA, 0.4 * 2 μL of target primers, 7.2 μL of water, and 10 μL of SYBR Fast Master Mix. Hot-start PCR was performed for 45 cycles, with each cycle consisting of denaturation for 5 s at 94 °C, annealing for 15 s at 58 °C, and elongation for 10 s at 72 °C. The CFX manager software (version 2.0, Roche Bio-Rad, USA) was used for data analysis. Relative quantification was done using the 2^−∆∆CT^ method [46]. Expression was normalized against the housekeeping gene β-globin. The primers used are shown in Table 1. 

### 2.6. Western Blot Analysis

An equal amount of protein (30 mg) from cell lysate was loaded into each well of a 12% SDS-polyacrylamide gel after denaturation with SDS loading buffer. After electrophoresis, proteins were transferred to a PVDF membrane, incubated with blocking buffer (5% fat-free milk) for 1 h at room temperature, and blotted with the following antibodies overnight: Anti-GAPDH (Cell Signaling Technology) and other antibodies (Abcam). The membrane was incubated with horseradish peroxidase-conjugated secondary antibodies for 1 h at room temperature. All signals were visualized and analyzed by Clinx ChemiCapture software (Clinx, Shanghai, China)

### 2.7. Analysis of Proteomic

Proteomic analysis was performed by Beijing Qinglian Baiao Biotechnology Co., Ltd.

MaxQuant software was used to process the MS raw data. The MS data were searched against the cabbage genome database (http://www.ocri-genomics.org/bolbase/). Parameters of MaxQuant searches referred to predecessors’ practices [47]. In order to extract the quantified information of the whole replica, the identification transfer protocol (“match-between-runs” feature in MaxQuant) was implemented in the experimental replica [48].

Quantitative data acquisition methods for all peptides in samples also referred to previous practices. The peak strength of the whole set of measurements was compared with Perseus (Version 1.4.1.3) [47].

### 2.8. Analysis of the Whole Transcriptome

Transcriptome analysis was performed by Beijing Zhongke Jingyun Technology Co., Ltd.

#### 2.8.1. Transcriptions of lncRNA and circRNA

In total, 1 ug of total RNA extracted from each BAT tissue was used for both mRNA and circRNA library construction. First, the total RNA was subjected to poly-d(A)-RNA isolation using NEBNext Magnetic Oligo d(T)25 Beads (NEB, S1419). The isolated poly-d(A)-RNA was used for mRNA library preparation using an NEBNext UltraTM RNA Library Prep Kit for Illumina (NEB, E7530) according to the instruction manual while the remaining RNA was further utilized for circRNA library preparation. After rRNA depletion using an NEBNext rRNA Depletion Kit (NEB, E6310) and linear RNA digestion with RNase R (epicenter, RNR07250), the circRNA was ethanol precipitated and reverse transcripted using random primers. Purified cDNA was then used for library construction using a KAPA Hyper Prep Kit (KAPA Biosystems, KK8504).

All of the RNA-seq libraries were subjected to 150 bp pair-end sequencing on an Illumina HiSeq X Ten platform. After sequencing, all reads passed through the filter were trimmed to remove low-quality bases and adaptor sequences. Reads then were aligned to the mm10 reference genome using tophat2 (v2.0.13). The results from mRNA-seq were also utilized for lncRNA analysis, which could reveal the poly-d(A)-lncRNA information.

For mRNA analysis, FPKMs were calculated and normalized using cufflinks (v2.2.1). The differentially expressed genes were calculated using the default parameter of cuffdiff (v2.2.1). Hierarchical clustering was carried out on log2(FPKM+1) across samples. Genes used for clustering were selected by maximum FPKM ≥ 1 and with a top 10% standard deviation of log2(FPKM + 1). The differentially expressed genes (DEGs) were further analyzed based on GO biological processes, molecular functions, and the KEGG pathway.

The lncRNA screening, function prediction, and novel lncRNA identification were achieved based on an lncRNA calling protocol [49] and the CNCI program [50]. The differentially expressed lncRNAs between samples were further calculated using cuffdiff (v2.2.1). Their corresponding target genes further undergwent GO enrichment and KEGG analysis.

#### 2.8.2. Transcriptions of microRNA

In total, 1 ug of total RNA extracted from each BAT tissue was used for miRNA library construction using an NEBNext Small RNA Library Prep Set for Illumina (NEB, E7330) according to the instruction manual. Briefly, the 3′ SR adaptor was first ligated followed by hybridization of the reverse transcription primer. The reverse transcription was performed after the ligation of 5′ SR adaptor, and then followed by 12 cycles of PCR enrichments (94 °C 30 s; 94 °C 15 s, 62 °C 30 s, 70 °C 15 s for 12 cycles; 70 °C 5 min). After amplification, ~140 bp DNA fragments were size-selected and recovered after resolving on the 6% PAGE gel.

The miRNA libraries were then sequenced on an Illumina HiSeq 2500 platform with 50 bp single-end read. After sequencing, all reads passed through the filter were trimmed to remove low-quality bases and adaptor sequences. Reads were then aligned with miRBase (miRBase20, http://www.mirbase.org/) of a mouse using Bowtie2 (version 2.3.4.1). Identification of the precursors of miRNA and prediction of novel miRNAs were achieved using the miRDeep2 (version 0.0.2). The miRanda tools (version 3.3a) were further used for the prediction of miRNA targets and their target genes were further analyzed based on GO and the KEGG database.

#### 2.8.3. Transcriptions of DNA Methylation

In total, 100 ng genomic DNA extracted from each BAT tissue were used for RRBS library construction following the protocol of Guo et al. [51]. The brief procedure included MspI digestion, end repair/dA tailing, and adapter ligation, which were accomplished by adding the corresponding reaction components sequentially and inactivating the enzymes by heating. Next, the ligation was performed by overnight incubation using the premethylated sequencing adapters and highly concentrated T4 DNA ligase. The ligated DNA fragments were directly processed until bisulfite conversion, and after this step, the DNA was purified. Consequently, the DNA was PCR-amplified and the fragments between 200 and 700 bp were gel-selected and purified for sequencing. Then, 150-bp pair-end sequencing was performed on an Illumina HiSeq X Ten platform.

After sequencing, raw reads were trimmed to remove low-quality bases and adaptor sequences and were mapped to the mouse reference genome (mm10, downloaded from the UCSC genome browser) using BS-Seeker2 (v2.1.1). Additionally, the lambda DNA genome was rebuilt as an extra reference for later calculation of the bisulfite conversion rate of each sample. The bulk methylation levels were analyzed for cytosine sites with ≥10X in each sample using CGmapTools [52]. A heatmap showing the average methylation levels of CG in bins across whole genomes (bin size, 500,000 bp) was generated by CGmapTools. Furthermore, differentially methylated region (DMR) analysis was carried out by CGmapTools using the dynamic region strategy. The differentially methylated regions related genes (DMGs) were further analyzed based on GO biological processes and molecular functions using the PANTHER classification system (http://www.pantherdb.org/).

### 2.9. Statistics

All of the qPCR results used a single-factor analysis of variance (ANOVA) followed by a two-tailed Student’s *t*-test for comparisons. All qPCR data are presented as means ± SEM. Significant differences were considered when *p* < 0.05. Graph-Pad Prism7 (GraphPad Software, San Diego, CA, USA) was used for data analysis.

## 3. Results

### 3.1. The Morphology of aBAT and neBAT

In order to understand the differences in the morphology of aBAT and neBAT, we first examined the internal and surface morphology through H&E staining, TEM, and SEM. We clearly observed that adipocytes in neBAT are denser and smaller (Figure 1A). Interestingly, neBAT contains small lipid droplets of a relatively uniform size compared with aBAT (Figure 1A). However, the adipocytes in aBAT were larger and the size of the lipid droplets varied from small to large lipid droplets. It can be seen from the droplet particle size distribution diagram (Figure 1A) that the particle size distribution of neBAT is 2 to 14 μm while that of aBAT is 10 to 40 μm. The lipid droplet size of neBAT is obviously smaller than that of aBAT (*p* < 0.0001). Furthermore, in order to further understand the dynamic changes during BAT development after delivery, especially the number of mitochondria; we observed the internal morphology of BAT from 0 to 8 weeks by TEM (Figure 1B). We found that neBAT had smaller adipocytes with smaller and less numerous lipid droplets but with a higher density of mitochondria (Figure 1B). Adipocytes, with the increased volume of lipid droplets in BAT, grew larger during post-delivery development (Figure 1B). Correspondingly, the number of mitochondria in BAT gradually decreased during post-delivery development. Interestingly, the volume of mitochondria increased in aBAT in comparison to neBAT (Figure 1B). This result is further shown in the particle size distribution map (Figure 1B). In the same area selected randomly, there were 45 mitochondria in neBAT but only 6 mitochondria in 8-week-old BAT. The size distribution of mitochondria in the BAT of newborn mice was 0.1 to 0.8 μm while with the increase of the age and size distribution of mitochondria in BAT of 8-week-old mice was 1.5 to 4 μm. The mitochondrial size of BAT in neBAT (W0) was dramatically smaller than that in aBAT(W8) (*p* < 0.0001). These changes may be compatible with the energy intake and utilization during post-delivery development. 

### 3.2. Brown Adipogenesis and Activation during Post-Delivery Development

To explore the features of brown adipogenesis and activation in BAT during post-delivery development, we performed the quantitative real-time PCR (qRTPCR) and western blot (WB). First, we examined the expression of *UCP1*, the critical thermogenic element in BAT. As expected, the expression of *UCP1* gradually decreased with increasing age (Figure 2A), which is consistent with the above morphological features. Interestingly, other thermogenic genes, including *PRDM16*, *PGC1α*, *PGC1β*, and *CPT1α*, showed a tendency to increase in week 2 and 4 before decreasing in week 6 and 8 while *PPARα* expression decreased during post-delivery development (Figure 2C). In addition, the mitochondrial genes (*Tfam* and *NRF1*) were also downregulated during post-delivery development (Figure 2D). Furthermore, we found that the expression levels of brown adipogenic genes, *PPARγ* and *CEBPα*, decreased and were significantly lower in W8-BAT (8 weeks after delivery) than W0-BAT (neonatal BAT) (Figure 2E), whereas the adipocyte differentiation-related genes (*CEBPβ* and *CEBPδ*) were upregulated (Figure 2E). The expression of lipid synthesis genes, including *FASN*, *ACC*, and *FABP4*, gradually increased during post-delivery development (Figure 2F). We further characterized the mRNA expression levels of *EN1* and *Myf5* by qRTPCR and observed a decreasing expression trend (Figure 2G). This may be due to the presence of more progenitors in the BAT of newborns, which may be one of the reasons why we extracted brown primary adipocytes from newborn mice. Consistent with the mRNA expression, the protein expression levels of *UCP1* and OXPHOS, including *ATP5A*, *UQCRC2*, *MTCO1*, *SDHB*, and *NDUFB8*, in neBAT and aBAT, were significantly higher in neBAT compared with aBAT (Figure 2H). This indicates that neBATs are more active than aBAT, and the activity and function in BAT maintain a decreasing trend during postnatal development except the beginning of the puberty (in week 2 and/or week 4 after delivery, the expression level of some thermogenic genes increased).

### 3.3. NeBATs Are More Active Than Embryonic BAT in Mice

To further study the function and molecular characteristics of neBAT, we also examined the activation and function of BAT in the embryonic period of mice (E19–20) (ME-BAT) and neBAT at the molecular level (Figure 3). We found that the protein expression of *UCP1* and OXPHOS in BAT was significantly upregulated in neBAT (day 1 and day 3 after delivery) compared to ME-BAT (Figure 3A); this may be due to the differences in the substrates of energy intake and consumption in neonatal mice. Correspondingly, the expression levels of brown adipogenic genes, thermogenic genes, and mitochondrial genes were all upregulated in neBAT compared with ME-BAT (Figure 3B–D). These may be adaptive changes due to cold stimulation and breastfeeding after birth, which also contributes to the maintenance of the body temperature during the fetal period. Interestingly, we also found that the expression levels of *Myf5* and *EN1* were significantly downregulated in neBAT compared with ME-BAT (Figure 3E). Combined with the results from the previous section, these findings suggest that the activity of neBAT was not only higher than aBAT but also higher than ME-BAT. However, the progenitor markers (Myf5 and EN1) in BATs gradually decreased during development from embryonic to adult. Further investigations on the dynamic process of BAT from stem cells to mature BAT cells are warranted.

### 3.4. Analysis of Proteomic Data in neBAT and aBAT

In order to further understand the difference between the neBAT and aBAT, we analyzed the features of neBAT and aBAT by proteomics. A total of 3071 proteins were identified, showing a good reproducibility based on the heat map (Figure 4A). Differentially expressed proteins (DEPs) are represented by a heat map (Figure 4B) and volcano map (Figure 4C) containing 911 downregulated proteins and 482 upregulated proteins by a comparative analysis of the aBAT to neBAT ratio (Figure 4D). The distribution of proteins between neBAT and aBAT are shown by the Venn diagram (Figure 4E). Among these proteins, a total of 2272 co-differential proteins (expression in both neBATs and aBATs) and 594 specific expression proteins in neBAT in contrast to 205 specific expression proteins in aBAT were found (Figure 3E). The distribution of DEPs determined by PCA (principal component analysis) showed obvious differences (Figure 4F). To reveal the annotations and classification of DEPs, gene ontology (GO) enrichment analysis was performed and presented in three forms, including the biological process (BP), cellular component (CC), and molecular function (MF). In the CC category, enriched GO terms were mainly associated with the ribosome, mitochondrion, plasma membrane, and membrane (Figure 4G). In the MF category, GO terms enriched for DEPs in neBAT and aBAT included rRNA binding, signal transducer activity, metal ion binding, and receptor binding (Figure 4H). The most frequent GO terms of BP were associated with signal transduction, vesicle-mediated transport, cell motility, and cell cycle (Figure 4I). Notably, the top four terms in the three categories (CC, MF, and BP) almost kept a higher countdown number in aBAT. To some extent, these results indicate that neBAT may maintain stronger biological activity in the newborn period. To reveal the significant enrichment of DEPs in the pathway terms, we performed pathway annotation of DEPs though the KEGG database. The DEPs were enriched mainly in the pathways of ribosomes, spliceosome, biosynthesis of antibiotics, RNA degradation, pyruvate metabolism, fatty acid biosynthesis, propanoate metabolism, and citrate cycle (TCA cycle) (Figure 4J). Consistently, we can clearly see that the top four terms, but not the biosynthesis of antibiotics, also possess a higher countdown number in aBAT to neBAT (Figure 4J).

### 3.5. Comprehensive Analysis of Whole Transcriptome Data in neBAT and aBAT

To further explore the features of BATs during post-delivery development, the characteristics and differences between aBATs and neBATs were comprehensively analyzed by whole transcriptome analysis, including mRNAs, microRNAs, lncRNAs, circRNAs, and DNA methylation.

First, we analyzed mRNA expression levels. Differentially expressed genes (DEGs) are represented by a heat map (Figure 5A) and volcano map (Figure 5B), which contain 1823 downregulated genes and 1821 upregulated genes by a comparative analysis of neBAT and aBAT (Figure 5C). The distribution of genes between neBATs and aBATs is shown in the Venn diagram (Figure 5D). Among these genes, a total of 3645 co-differential genes and 231 specific expression genes in neBAT as well as 22 specific expression genes in aBAT were identified (Figure 5D). GO enrichment analysis was performed, and in the CC, DEGs in the neBAT and aBAT category were mainly associated with the cytoplasm, membrane, and mitochondrion (Figure 5E). In the MF category, GO terms enriched for DEGs in neBAT and aBAT included protein binding, identical protein binding, and RNA binding (Figure 5F). The most enriched GO terms of BP were associated with translation (Figure 5G). To reveal the significant enrichment of DEPs in pathway terms, we performed pathway annotation of DEGs through the KEGG database. The DEGs were enriched mainly in the metabolic pathways, cancer pathways, ribosome, and PI3K–Akt signaling pathways (Figure 5H). This result is compatible with the pathway analysis of the proteome, indicating that the genes involved in metabolism and differentiation are active in neBATs compared with aBATs.

DNA methylation is an essential modification for the regulation of gene expression and cessation, which is involved in many diseases, such as cancer, aging, and Alzheimer’s disease. DNA methylation is one of the important components of epigenetics. In this study, we identified a total of 50,133 differentially methylated regions (DMRs) in neBAT and aBAT, which were mostly localized to the CG site (Figure 6A,B). The heat map of the methylation of the CG site (mCG) shows a high abundance of methylation regions in neBAT (43.7%) compared with aBAT (38.6%) (Figure 6C). Furthermore, we identified 1247 hyper- and 1667 hypo-DMRs while 932 hyper- and 1041 hypo-DMRs were related to promoters by differentially methylated region analysis (Figure 6D). Furthermore, compared with aBATs, we found 383 hyper- and 503 hypo-DMR-related genes in neBATs by DMR analysis. Although we found that methylation modification is more abundant in neBAT compared with aBAT, the level of hyper-DMRs is indeed lower (Figure 6C,D). This event was further verified at the more intuitive methylation levels in bins (Figure 6). As we can observe in the figures, all of the methylation levels of the CG, CHG, and CHH sites located on different chromosomes show a high level in aBAT compared with neBAT (Figure 6E). These results suggest that more gene activity may be inhibited by methylation in aBAT than in neBAT, which may indicate that neBATs are in a more biologically dynamic state than aBAT.

In recent years, CircRNA, a new class of single-stranded and covalently closed circular RNA, which aroused widespread concern among researchers, could regulate the activity of miRNAs as efficient miRNA sponges and play an important regulatory role in some diseases by interacting with miRNAs [53]. In our study, we thoroughly analyzed the expression characteristics of circRNA in neBAT and aBAT. The filter circRNA and candidate circRNA in each tissue of neBAT and aBAT are shown in Appendix A. Interestingly, we found that there are more novel circRNAs compared with overlapped circRNAs as identified in circBase when analyzed in neBAT and aBAT (Figure 7C). The length of the most exonic circRNAs was less than 1500 nucleotide (nt), and the median length was about 400 nt (Appendix A). About 60% of the circRNAs consisted of exons, whereas smaller fractions aligned with introns and intergenic spacers (Figure 7E). Moreover, there are 944 circRNAs that were explicitly expressed in neBAT, and 1361 circRNA that were specifically expressed in neBAT. By contrast, only 107 circRNAs overlapped in neBAT and aBAT (Figure 7B,C).

In order to explore the feature of the parental gene for circRNAs, gene ontology and KEGG pathway analysis of the host genes of differently expressed circRNAs was performed (Figure 7F,G). In the BP category, enriched GO terms were mainly associated with positive regulation of the phosphorus metabolic process, muscle cell differentiation, and regulation of the cell cycle process (Figure 7F). In the CC category, GO terms enriched for host genes in neBAT and aBAT included the cytoplasm, ESC/E (Z) complexes, pronucleus, PcG protein complexes, and intracellular parts (Figure 7F. Moreover, most GO terms of MF were associated with enzyme activator activity, carbohydrate response element binding, protein heterodimerization activity, and primary miRNA binding (Figure 7F). Regarding pathway terms, the host genes were enriched mainly in renal cell carcinoma, prostate cancer, phospholipase D signaling pathway, microRNA in cancer, melanoma, and glycerolipid metabolism (Figure 7G). 

We also showed a solicitude for the differences between microRNA and LncRNA between neBAT and aBAT. The transcriptome of microRNAs contained a total of 486 microRNAs and identified 236 differentially expressed microRNAs between neBATs and aBATs, including 90 upregulated and 146 downregulated microRNAs by filtering analysis of fold change ≥2.0, *p* < 0.05, and FDR < 0.05 (Appendix A). Meanwhile, a total of 5321 lncRNAs were detected, whereas 271 lncRNAs appeared to be differentially expressed, with fold changes ≥2.0, *p* < 0.05, and FDR < 0.05. Among these, 168 and 102 lncRNAs were upregulated and downregulated, respectively (Appendix A). More information on lncRNAs and microRNAs can be found in the Appendix A. By performing GO and KEGG analyses of the target genes of microRNAs and/or lncRNAs, it was indicated that the biological processes of these two types of non-coding RNA were related to specific cellular and metabolic processes (Appendix A), whereas the pathways were mainly related to cancer (Appendix A). This finding was based on the analysis of the top 200 differentially expressed lncRNA as shown in the heat map (Appendix A), in addition to the top 20 most enriched KEGG pathways (Appendix A).

### 3.6. Interaction among Transcriptome Factors

This study identified major differences between neBATs and aBATs at the RNA level. To further explore the relationship between these transcriptome factors, we pioneered a ceRNA network using our transcriptome data, including the differential expression elements of mRNA, miRNA, and circRNA in neBAT and aBAT (Figure 8A). We selected differentially expressed *mmu-circ-0000866* and *chr5_135597375_135599887*, sharing a common binding site of microRNA response elements and interacting with mRNA, including *dnajc28*, *ldhd*, and *svip* (Figure 8A).

Furthermore, we pioneered an interaction network, including mRNA, circRNA, and methylation, all of which are differentially expressed (Figure 8B). From the obtained visual network diagram, we can identify that circRNA, including *mmu-circ-0000866*, *chr5_135597375_135599887*, and *mmu_circ0001447*, participate in the regulation of mRNA based on the methylation level, including *trip6*, *clasp*, *bub1b*, and *accsl* (Figure 8B). These RNA interactions may provide a novel mechanism for the development and/or activation mechanisms of BAT during postnatal development. We also constructed a regulatory network, which includes the differential expression elements of circRNA, lncRNA, and mRNA in neBAT and aBAT (Figure 8C), as well as a more complex regulatory network that involved all of the differential expression elements in this study, including circRNA, lncRNA, microRNA, mRNA, transcription factors, and DNA methylation (Figure 9). Co-expression regulation analysis showed that some lncRNAs are located at the core of the interaction network and regulate the expression of numerous mRNAs. For example, *lnc-NONMMUG024827* regulates the expression of genes that are related to glucose and fatty acid metabolism genes, including *Adipoq* and *Acadsb*. Compared to neBATs, the expression of these two genes in the aBATs was downregulated. These results indicate that some pathways related to glucose and fatty acid metabolism may be more active in neBATs, which was mutually confirmed by RNA-Seq analysis (Figure 5H). Although the development and function of BAT has a very complicated regulatory network, this work provides a preliminary exploration.

Collectively, these results provide a foundation for a better understanding of BAT function and development. Our findings provide novel perspectives on neBATs and provide a basis for future research of the potential roles of neBATs, and show that the activity of aBATs is similar to that of neBATs due to epigenetic modifications.

## 4. Discussion

In recent years, BAT has received a high degree of attention because of the potential therapeutic effects on obesity and related metabolic diseases [54]. However, the characteristics and function of neBAT are much less understood, and the specific differences between neonatal BAT and adult BAT have not been clearly explored. In this study, we provided evidence that neBATs are more dynamic BAT than aBAT through a comprehensive analysis from morphology, molecular, and omics perspectives. Despite these findings, there are still many details awaiting further research.

We comprehensively revealed the internal and surface morphology of neBAT though H&E staining, TEM, and SEM. Compared with aBAT, neBAT shows bright features that contain smaller lipid droplets (Figure 1A) and more mitochondria (Figure 1B), which is compatible with the known function of neBAT in the neonatal period. As is known, mitochondria are the major sources of heat and oxidative phosphorylation in BAT. To further understand the mitochondria changes in BAT, we detected the morphological and molecular features of BAT by TEM (Figure 1B), RT-PCR, and western blot. (Figure 2). We demonstrated that the number of mitochondria decreases (Figure 1B) and the expression level of *UCP1* also gradually decreases during post-delivery development (Figure 2A). Interestingly, the mitochondria volume in BAT increases during post-delivery development (Figure 1B). Based on this discovery, we hypothesize that there may be an adaptive developmental process in mitochondria. Specific differences in mitochondria between neBAT and aBAT remain to be further studied.

In addition, previous studies on post-delivery BAT were limited to phenotypic studies. We first revealed the dynamic changes of molecular features of BAT during post-delivery development. As a result, the brown adipogenic genes, thermogenic genes, and mitochondrial genes were significantly upregulated in neBAT compared with aBAT. Lipid synthesis genes demonstrated significantly elevated expression in aBAT. Notably, the expression of adipocyte differentiation genes (*C/EBPβ* and *C/EBPδ*) also gradually increased during post-delivery development (Figure 2D). *C/EBPβ* and *C/EBPδ* are crucial for adipogenesis, which has been confirmed by a double-knockout *C/EBPβ* and *C/EBPδ* mice study with a further decline in adipose tissue mass [55]. Some studies indicate that the effect of *C/EBPβ* is less evident in embryonic fibroblasts [9], which may explain the reason for the minimal expression in neBAT compared with aBAT. In addition, the expression levels of thermogenic genes, including *PGC1α*, *PGC1β*, and *CPT1α* (*p* = 0.075), and mitochondrial genes, *Tfam* and *NRF1*, were increased in week 2 and/or 4 after delivery (Figure 2C). This may be related to adolescent development [56,57] or breastfeeding. Furthermore, we incidentally observed that the expression levels of *Myf5* and *EN1* gradually decreased during post-delivery development, and there was a significant difference between neBAT and aBAT. This suggests a decrease of progenitor cells in BAT during post-delivery development. Finally, we also examined the activation and function of BAT in the embryonic period and the first few days after delivery (Appendix A). The protein expressions of *UCP1* and *OXPHOS* were significantly upregulated in neBAT compared with ME-BAT (Appendix A). Notably, *Myf5* and *EN1* were significantly downregulated in neBAT compared with MEBAT. This further validates our hypothesis that the amount of progenitor cells in BAT would gradually decrease from the embryonic period to adulthood. Combining all the results above, we can conclude that neonatal BAT may be the most active and functional BAT.

The results of proteomics further reveal the differences between neBATs and aBATs. Compared with neBAT, there were more downregulated DEPs than upregulated DEPs in aBAT. The enrichment analysis of GO also reveals an attractive phenomenon: The functional categories of cellular components (ribosome, mitochondrion, plasma membrane, and membrane), molecular function (rRNA binding, signal transducer activity, metal ion binding, and receptor binding), and biological processes (signal transduction, vesicle-mediated transport, cell motility, and cell cycle) (Figure 3G–J) were all enriched in aBATs compared to neBATs. These phenomena may reveal the reason why neBATs are more active than aBATs at the protein level: Mitochondrial activity, protein synthesis, and cell life activity levels were higher in neBAT. Consistently, KEGG analysis showed that the pathway of ribosomes possessed the highest countdown number in aBAT compared to neBAT (Figure 3J).

To further explore the features of neBAT and aBAT, the characteristics and differences of each were comprehensively analyzed using whole transcriptome data, including mRNA, microRNA, lncRNA, circRNA, and DNA methylation. Thousands of significantly different lncRNAs and mRNAs were identified in neBAT and aBAT. Furthermore, we found 383 hyper- and 503 hypo-DMR-related genes in aBATs by DMR analysis compared to neBAT. Importantly, we found 1221 new circRNA in neBATs and 1991 new circRNAs in aBAT. Using full transcriptome analysis, we thoroughly explored the characteristics and differences of neBAT and aBAT on a transcriptome level, and obtained many interesting findings. In particular, an analysis of the circular RNA and DNA methylation showed a more intuitive difference between neBAT and aBAT (Figure 5 and Figure 6). These differences indicate that both circular RNA and DNA methylation are involved in the post-delivery developmental regulation of BAT and further confirmed that neBATs are more active/dynamic compared with aBAT.

To further analyze the biological significance of these DEGs/DEPs, enrichment analysis of KEGG was performed, which revealed many key metabolic pathways related to BAT’s dynamic changes during post-delivery development. Among the top 20 enriched KEGG pathways (Figure 4H), 24 upregulated DEGs that are related to glycolysis/gluconeogenesis were isolated, suggesting that the glycolysis/gluconeogenesis pathway was activated for ATP production for neBAT. This phenomenon is consistent with the results of previous morphological and molecular experiments, indicating that neBATs have a stronger ability to provide energy compared to aBAT. Compared with glycolysis/gluconeogenesis pathway, the BCAA (valine, leucine, and isoleucine) oxidation pathway and fatty acid metabolism pathways have higher enrichment (‘rich factor’ in Figure 4H), with 25 and 22 DEGs that were upregulated, respectively. Previous studies have shown that BCAA catabolism in BATs controls energy homeostasis, and intracellular triglycerides are the primary energy source for cold-induced interscapular BAT thermogenesis [58,59]. Thus, we hypothesize that compared with glucose, fatty acids and amino acids are more likely as the substrates of BAT thermogenics, which needs to be further verified using isotope labeling and metabolomics analysis.

To identify the key factors that lead to such a dramatic difference between neBAT and aBAT, we mapped the interaction network of various types of RNAs (Figure 8 and Figure 9). Based on this interaction network, we found that mRNA *Dnajc28*, which is a member of the heat shock protein family, as well as circRNA *Chr5_135597375_135599887*, one of genes that located on chromosome 5, were in the center of the interaction network, and involved in the regulation of extensive genes related to miRNA. This discovery strongly suggests that these two genes may play a crucial role in regulating BAT function and activity. Furthermore, many interactions between lncRNAs and mRNAs have also been revealed (Figure 8C). Adiponectin, an endogenous bioactive polypeptide secreted by fat cells, is closely related to insulin sensitivity, glucose metabolism, and fat metabolism [60]. Its mRNA expression is upregulated by lncRNA NONMMUG024827 in neBATs. This phenomenon indicates that some pathways of glucose and fatty acid metabolism may be more active in neBAT. Another downstream lncRNA at the core of the interaction network is NONMMUG004953, and the expression of some mRNAs related to T cell function (*Mr1*) [61] and inflammation (*NAAA*) [62] was altered. There are three lncRNAs located close together in the interaction network: NONMMUG034109, NONMMUG021221, and NONMMUG008287; these are involved in the regulation of the mRNA expression of a large number of cancer-related genes, such as *Pik3r1*, *Gulp1*, *Chrdl1*, and *Crebaf* [63,64,65,66]. These phenomena suggest that BAT may be closely related to cancer, which coincides with the results of the RNA-Seq (Appendix A), and may provide an extremely novel approach for the treatment of cancer. Collectively, based on the preliminary analysis of the interaction network, BATs may be related to glucose metabolism, lipid metabolism, immunity, and cancer.

The development and function of BATs may have a very complicated regulatory network, and this study provides only a preliminary exploration. However, our comprehensive study has revealed the differences between neBATs and aBATs at the morphological, molecular, and omics levels, which lays a foundation for improving BAT function and activity through gene regulation or epigenetic regulation.

## 5. Conclusions

In summary, we investigated the dynamic changes during post-delivery development at the morphological, molecular, and omics levels. Our results confirm that newborn BAT is more active than adult BAT. Collectively, compared with aBATs, neBATs have better mitochondrial and thermogenic functions, which may be closely related to the glycolysis/gluconeogenesis, BCAA oxidation, and fatty acid metabolism pathways. At the genetic level, BATs may be related to the expression of genes related to glucose metabolism, lipid metabolism, immunity, and cancer. Our research can serve as a guide for establishing methods in transforming the less active adult BATs into the more active and younger newborn BATs, or improving the functional activity of BATs through gene regulation or epigenetic regulation.

## Figures and Tables

**Figure 1 cells-09-00201-f001:**
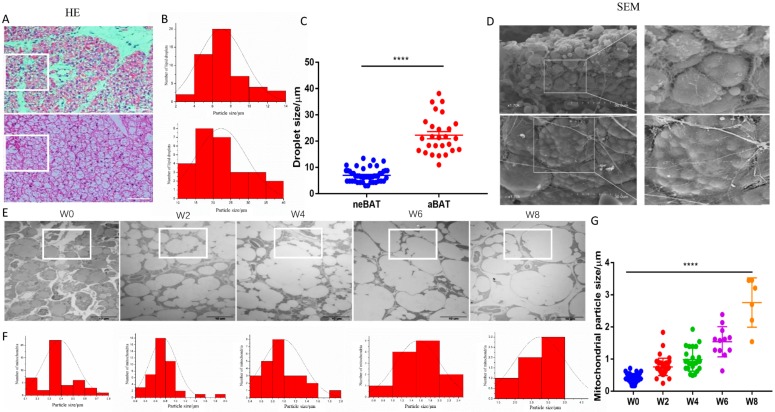
The morphology of aBAT and neBAT. (**A**) H&E staining (scale bars, 100 μm) of neBAT and aBAT. (**B**,**C**) The size distribution of the lipid droplets is shown in the white box of the figure (**A**). (**D**) SEM (scale bars, 30 μm) of neBAT and aBAT (the second right-hand image is a local enlargement of the second image on the right). (**E**) TEM of BAT on week 0 (W0), week 2 (W2), week 4 (W4), week 6 (W6), and week 8 (W8) after pregnancy. The first row is 3000 × (scale bars, 10 μm). (**F**,**G**) The mitochondrial particle size distribution in the white box of the figure above, respectively. Data are presented as the mean ± sem. The significance of the difference was set at **** *p* < 0.0001 relative to W0.

**Figure 2 cells-09-00201-f002:**
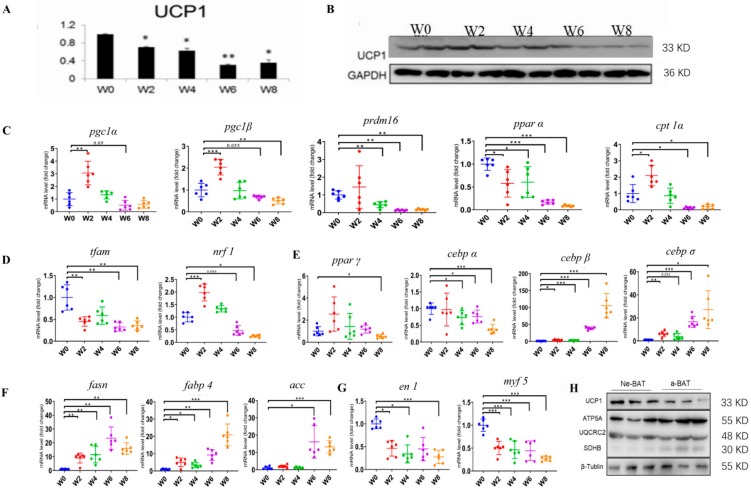
The molecular dynamic changes in BATs during development post-pregnancy. (**A**) mRNA and (**B**) protein expression levels of *ucp1*, (**C**) mRNA expression levels of other thermogenic genes (*pgc1α*, *pgc1β*, *prdm16*, *pparα*, and *cpt1α*), (**D**) mitochondrial genes (*tfam*, *nrf1*), (**E**) brown adipogenic synthesis (*pparγ* and *cebpα*), and adipocyte differentiation genes (*cebpβ* and *cebpδ*), (**F**) lipid synthesis genes (*fasn*, *fabp4*, and *acc*), and (**G**) brown adipocyte progenitor markers (*myf5* and *en1*). (**H**) Protein expression levels of *ucp1* and *oxphos* (*atp5a*, *uqcrc2*, and *sdhb*). Data are presented as the mean ± sem. n = 6 per group. The significance of the difference was set at * *p* < 0.05, ** *p* < 0.01 and *** *p* < 0.001 relative to W0.

**Figure 3 cells-09-00201-f003:**
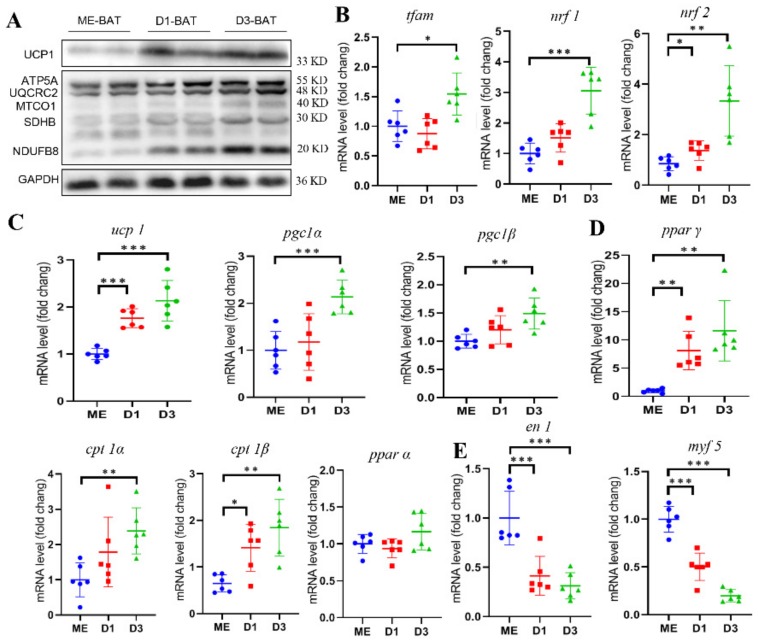
neBAT is more active than embryonic BAT in mice. (**A**) The protein expression levels of *ucp1* and *oxphos* (including *atp5a*, *uqcrc2*, *mtco1*, *sdhb*, and *ndufb8*. The molecular weight of each band is marked on the right) in neBAT (D1 and D3) and ME-BAT. (**B**) The mRNA expression levels of mitochondrial genes (including *tfam*, *nrf1*, and *nrf2*), (**C**) thermogenic genes (including *ucp1*, *pgc1α*, *pgc1β*, *cpt1α*, *cpt1β*, and *pparα*), (**D**) brown adipogenic synthesis *pparγ*, and (**E**) brown adipocyte progenitors (including *myf5* and *en1*). Data are presented as the mean ± sem n = 6 per group. The significance of the difference was set at * *p* < 0.05, ** *p* < 0.01, and *** *p* < 0.001 when compared to ME-BAT, respectively.

**Figure 4 cells-09-00201-f004:**
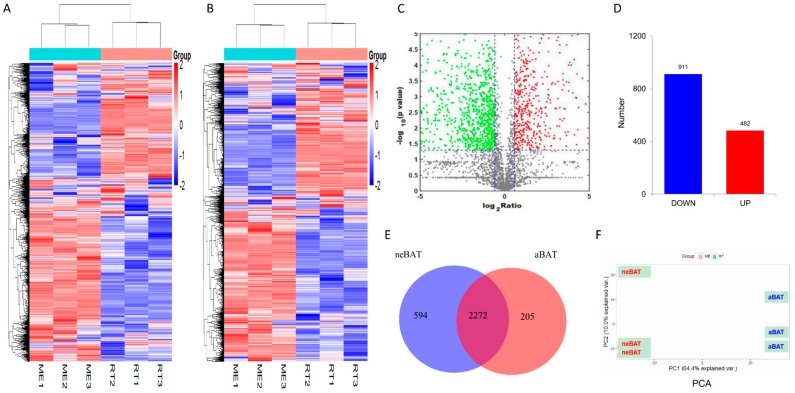
Analysis of proteomics data in neBATs and aBATs. (**A**) Heat map of the total proteins and (**B**) differentially expressed proteins (DEPs) with fold change >1.5 and *p* value <0.05 in neBAT and aBAT. (**C**) Volcano plot of the proteome. The volcano plots show significance on the y-axis (−log10, *p* value) against the protein expression ratio (log2, fold change of aBAT vs. neBAT). The FDR cutoff of <0.05 is indicated by the blue dashed horizontal lines. The red plots represent upregulation of proteins and the green plots indicate downregulation of proteins. (**D**) Number of proteins upregulated and downregulated in neBATs and aBATs. (**E**) Venn diagrams of neBAT vs. aBAT. (**F**) Principal component analysis (PCA) results are shown in red (neBAT) and blue (aBAT). (**G**–**I**) GO categories for DEPs in the proteome. (**J**) KEGG pathway analysis of DEPs. Numbers and *p*-values of DEPs in each pathway.

**Figure 5 cells-09-00201-f005:**
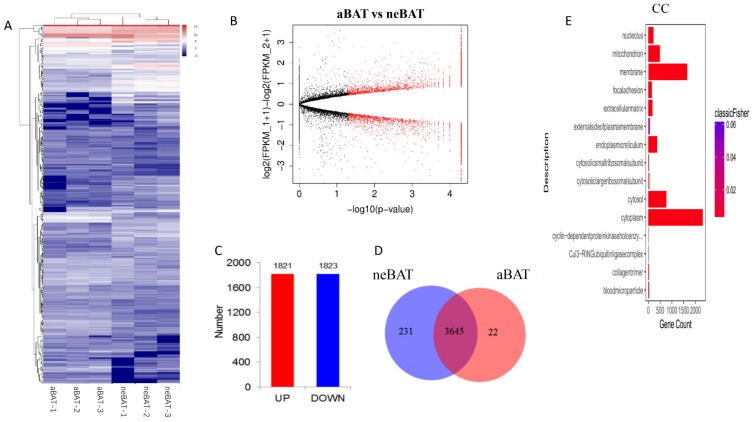
The analysis of mRNA-seq data from neBAT and aBAT. (**A**) Heat map showing expression profiles of mRNA. (**B**) Volcano map of mRNA. The volcano plot shows significance on the y-axis (−log10, *p*-value) against the gene expression ratio (log2, fold change of neBAT versus aBAT), and the FDR cutoff <0.05 is indicated. (**C**) Number of upregulated and downregulated genes in neBAT and aBAT. (**D**) Venn diagrams of neBAT versus aBAT. (**E**–**G**) GO categories for DEGs. (**H**) KEGG analysis of DEGs. Count number and *p*-value of DEGs in pathways.

**Figure 6 cells-09-00201-f006:**
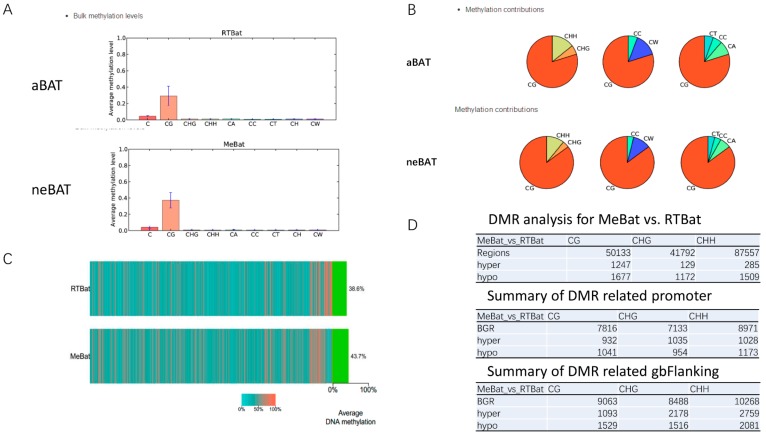
Analysis of DNA methylation patterns of aBATs and neBATs. (**A**) Bulk methylation levels in aBATs and neBATs. The bulk methylation levels were analyzed for cytosine sites with ≥10X in each sample using CGmapTools. (**B**) Methylation contributions in aBATs and neBATs. The methylation contributions indicate chances that an observed site could be in a specific context. This analysis was applied to cytosine sites with ≥10X in each sample using CGmapTools. H = {A, C, T} and W = {C, T}. (**C**) This heat map shows the average methylation levels of CG in bins across whole genomes. The color indicates the average methylation levels in bins across the whole genome (bin size: 5 Mb). Green bars on the right indicate the global average DNA methylation levels. (**D**) Differentially methylated region (DMR) analysis was conducted by CGmapTools using the dynamic region strategy. (**E**) Distribution of mCG, mCHG, and mCHH in bins for neBATs and aBATs.

**Figure 7 cells-09-00201-f007:**
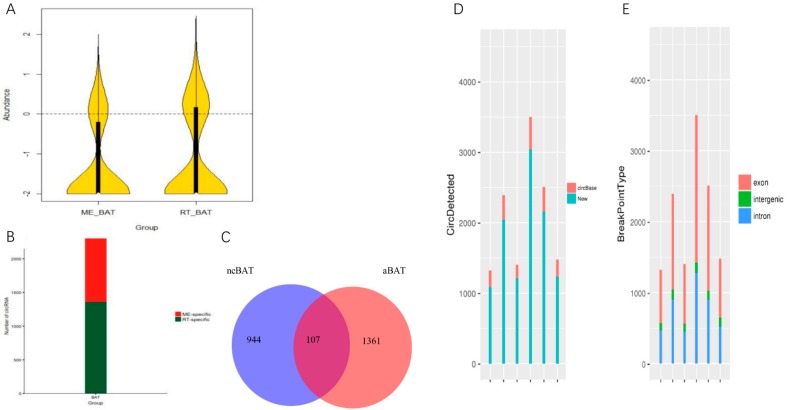
Analysis of circRNA data of neBAT and aBAT. (**A**) Violin plot of circRNA. The circRNA specifically expressed in each tissue was pooled and used in a Violin map. The vioplot shows the abundance on the y-axis. (**B**) Number of specific circRNAs in neBAT and aBAT. (**C**) Venn diagrams of neBAT versus aBAT. (**D**) New circRNAs and known circRNA in circBase. (**E**) Break point location for each circRNA. (**F**) GO categories for host genes of circRNAs. (**G**) KEGG analysis of host genes of circRNAs. Count number and *p*-value of host genes of circRNAs in pathways.

**Figure 8 cells-09-00201-f008:**
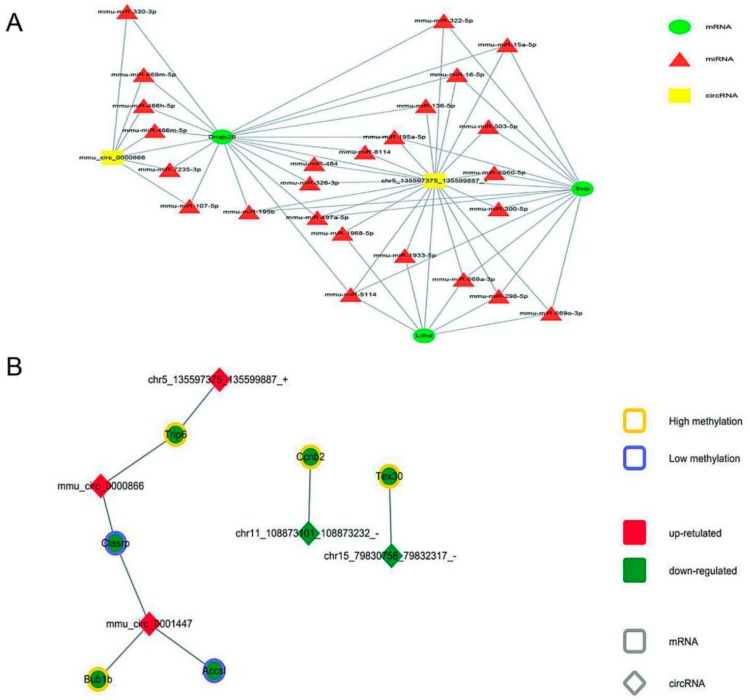
Competing endogenous RNA network (**A**) with mRNA, circRNA and DNA methylation regulation networks (**B**,**C**) in neBAT and aBAT. (**A**) The competing endogenous RNA network was based on miRNA, circRNA, and mRNA interactions. (**B**) The mRNA, circRNA, and DNA methylation regulation network was based on mRNA, circRNA, and DNA methylation interactions. (**C**) The regulation network of circRNA, lnc RNA, and mRNA in aBAT and neBAT.

**Figure 9 cells-09-00201-f009:**
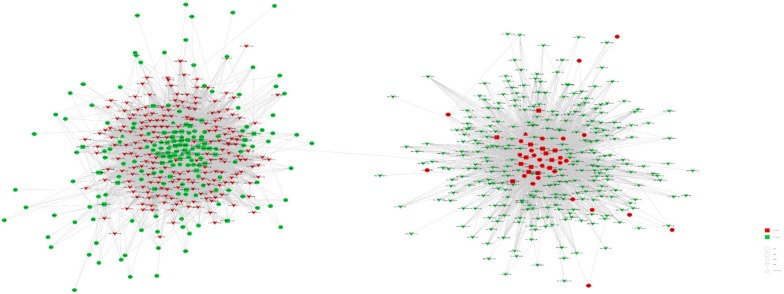
A complex regulatory network that involved all of the differential expression elements in this study, including circRNA, lncRNA, microRNA, mRNA, transcription factors, and DNA methylation.

**Table 1 cells-09-00201-t001:** Primer Sequences.

Primer	Forward (5′–3′)	Reverse (5′–3′)
ACC	AGCTGATCCTGCGAACCT	GCCAAGCGGATGTAAACT
AP2	GAAGACTGCGAGGACCTC	GAAGTGCCGTAATCCCCACAC
CEBP/α	GCGGGAACGCAACAACATC	GTCACTGGTCAACTCCAGCAC
CEBP/β	TGACGCAACACACGTGTAACTG	AACAACCCCGCAGGAACAT
CEBP/δ	CGACTTCAGCGCCTACATTGA	GAAGAGGTCGGCGAAGAGTT
CPT1α	GACTCCGCTCGCTCATTCC	GACTGTGAACTGGAAGGCCA
CyclophilinA	CAAATGCTGGACCAAACACA	GCCATCCAGCCATTCAGTCT
EN1	CTCACAGCAACCCCTAGTGT	CCGCTGCTCCGTGATATAG
Fasn	TAGAGGGAGCCAGAGAGACG	CCGACATACCGGCTATCACC
Myf5	GCCTTCGGAGCACACAAAG	TGACCTTCTTCAGGCGTCTAC
NRF1	CAACAGGGAAGAAACGGAAA	GCACCACATTCTCCAAAGGT
NRF2	TAGATGACCATGAGTCGCTTGC	GCCAAACTTGCTCCATGTCC
PGC1α	ACCGCTTTCTGGGTGGATT	TGAGGACCGCTAGCAAGTTT
PGC1β	CGTATTTGAGGACAGCAGCA	TACTGGGTGGGCTCTGGTAG
PPARα	AGCCTCAGCCAAGGTTGAACT	TGGGGAGAGAGGACAGATGG
PPARγ2	TCGCTGATGCACTGCCTATG	GAGAGGTCCACAGAGCTGATT
PRDM16	GAAGTCACAGGAGGACACGG	CTCGCTCCTCAACACACCTC
Tfam	GTCCATAGGCACCGTATTGC	CCCATGCTGGAAAAACACTT
UCP1	GGCAAAAACAGAAGGATTGC	TAAGCCGGCTGAGATCTTGT

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
