# Peer review of "Comprehensive Analysis of the Characteristics and Differences in Adult and Newborn Brown Adipose Tissue (BAT): Newborn BAT Is a More Active/Dynamic BAT"

_cells, 2020, doi:10.3390/cells9010201_

Round 1
Reviewer 1 Report
The manuscript submitted by Liu et al. investigated morphological features of adult brown (aBAT) fat and neonatal brown fat (neBAT) in mice. Authors showed neBAT contains less quantity and smaller size of lipid droplet, and higher mitochondrial content compared with aBAT. Authors used
Proteomic and transcriptomic analysis, and showed comprehensively the changes in neBAT and aBAT. Overall the study design and experiments well performed. Figures are poor quality, cannot read them. Figures must be improved and re submitted.
Author Response
Reviewer(s)' Comments to Author:
Reviewer: 1
<b>Comments and Suggestions for Authors<b>
The manuscript submitted by Liu et al. investigated morphological features of adult brown (aBAT) fat and neonatal brown fat (neBAT) in mice. Authors showed neBAT contains less quantity and smaller size of lipid droplet, and higher mitochondrial content compared with aBAT. Authors used
Proteomic and transcriptomic analysis, and showed comprehensively the changes in neBAT and aBAT. Overall the study design and experiments well performed. Figures are poor quality, cannot read them. Figures must be improved and re submitted.
Answer: Thank you very much for your professional comments
We have improved the figures and put all the figures in PDF (attachment)
Reviewer 2 Report
The manuscript from Liu J and colleagues describes a comparison between the brown adipose tissue (BAT) in adult and newborn and conclude that BAT from newborn is more active.
They analyse different features of BAT (aBAT vs neBAT). The logic they follow is not always clear to me; for example, circRNA are presented as a new class of RNA “which aroused widespread concern among researchers”…and thus they analysed also circRNA. I am sure there are valid scientific reasons to analyse circRNA, but this does not emerge from the text. It seems they performed this additional analysis just because it is “trendy”.
I appreciate the “omics” approaches they use, but it would be better to have some validations for the data presented. Moreover, it is not clear if the conclusions they draw are relevant to any kind of application. As minor points, I found some errors in the way of presenting western blots; typing errors are also present; quantifications are missing for the data shown in figure 1.
In conclusion, I think the data are not strong enough to be published at this level.
Author Response
Reviewer: 2
<b>Comments and Suggestions for Authors<b>
The manuscript from Liu J and colleagues describes a comparison between the brown adipose tissue (BAT) in adult and newborn and conclude that BAT from newborn is more active.
They analyse different features of BAT (aBAT vs neBAT). The logic they follow is not always clear to me; for example, circRNA are presented as a new class of RNA “which aroused widespread concern among researchers”…and thus they analysed also circRNA. I am sure there are valid scientific reasons to analyse circRNA, but this does not emerge from the text. It seems they performed this additional analysis just because it is “trendy”.
Answer: Thank you very much for your professional comments. CircRNA play an important regulatory role in some diseases by interacting with miRNAs associated with diseases. So the difference of circRNA between the two bats is of great significance. By mapping the gene interaction network, we found that there are indeed a large number of circRNA associated with the function of BAT, which will provide essential information for network regulation analysis. This phenomenon is also added to the ‘Discussion’.
I appreciate the “omics” approaches they use, but it would be better to have some validations for the data presented. Moreover, it is not clear if the conclusions they draw are relevant to any kind of application. As minor points, I found some errors in the way of presenting western blots; typing errors are also present; quantifications are missing for the data shown in figure 1.
In conclusion, I think the data are not strong enough to be published at this level.
Answer: we carefully reanalyzed the data. We have added an in-depth analysis of omics data to the discussion section of the article. About figure 1, we have marked the mitochondria of aBAT and neBAT with arrows in the figure 1B. It is obvious from the figure that the number of mitochondria gradually decreases with the growth time course, but the volume gradually increases.
Reviewer 3 Report
Liu J et al. compared morphological and molecular characteristics between neonatal and adult brown adipose tissue (BAT) by transmission/scanning electron microscopy, proteomics, and transcriptome analysis. The experiments were well performed, and this comprehensive study would be highly helpful for future studies. However, the analysis of the data should be done more carefully and in detail.
Major points
1) Lines: 245-248, 430-431: To conclude that enBAT is more thermogenic than in aBAT, the authors should directly measure oxygen consumption of the tissue specimens. Please refer a recent study that performed Seahorse assay ex vivo which can quantify oxygen consumption rate (OCR) in isolated tissues (Nat Med. 23: 1454-1465, 2017).
2) In case OCR measurement is difficult due to no availability of the Seahorse machine (or any other reason), the authors may alternatively analyze the certain substrate oxidative capacity using proteomics and/or transcriptome data. To this end, the authors should report fatty acid oxidation pathway, glucose oxidation pathway, and BCAA oxidation pathway all of which are known to be crucial for BAT thermogenic capacities (Nature 572: 614-619, 2019).
3) Lines 307-309: The authors indicated that metabolism-related pathways are upregulated in neBAT compared with aBAT. The authors should come clear what kind of metabolism is (or is not) upregulated in neBAT. Potential substrate preferences in neBAT (fatty acids vs glucose vs amino acids, etc) are highly interesting.
4) The data of present study is highly helpful for future studies; therefore, the protemic/trascriptome data should be uploaded in public database such as ProteomicsDB (https://www.proteomicsdb.org/) and ArrayExpress (https://www.ebi.ac.uk/arrayexpress/). To date, many of Journals requested authors to do so, because this will allow research communities to validate/re-analyze it, thereby increasing greatly the citation of the paper.
5) As labeling of figure is too small and difficult to read, all figures including supplemental figures must be revised. Please enlarge labeling so that readers can see easily without zooming up.
Minor points
6) Line 93: 1-3 days after pregnancy --> 1-3 days after “delivery” (if correct)
7) Line 94: (after pregnancy) --> (after “delivery”) (if correct)
8) Figure 1B: The authors concluded that volume of mitochondria increases in aBAT compared with neBAT (lines 221-222, 398-408). Please indicate mitochondria in Fig 1B by arrows and/or quantify the number of mitochondria per field.
Author Response
<b>Comments and Suggestions for Authors<b>
Liu J et al. compared morphological and molecular characteristics between neonatal and adult brown adipose tissue (BAT) by transmission/scanning electron microscopy, proteomics, and transcriptome analysis. The experiments were well performed, and this comprehensive study would be highly helpful for future studies. However, the analysis of the data should be done more carefully and in detail.
Major points
1) Lines: 245-248, 430-431: To conclude that enBAT is more thermogenic than in aBAT, the authors should directly measure oxygen consumption of the tissue specimens. Please refer a recent study that performed Seahorse assay ex vivo which can quantify oxygen consumption rate (OCR) in isolated tissues (Nat Med. 23: 1454-1465, 2017).
Answer: Thank you very much for your comments. Due to the limitations of the equipment conditions, we cannot perform this experiment. However, we did a complimentary analysis of the metabolism-related signaling pathway. Our answer attached under comment 3.
2)In case OCR measurement is difficult due to no availability of the Seahorse machine (or any other reason), the authors may alternatively analyze the certain substrate oxidative capacity using proteomics and/or transcriptome data. To this end, the authors should report fatty acid oxidation pathway, glucose oxidation pathway, and BCAA oxidation pathway all of which are known to be crucial for BAT thermogenic capacities (Nature 572: 614-619, 2019).
Answer: Thank you very much for your comments. Our answer attached under comment 3.
3) Lines 307-309: The authors indicated that metabolism-related pathways are upregulated in neBAT compared with aBAT. The authors should come clear what kind of metabolism is (or is not) upregulated in neBAT. Potential substrate preferences in neBAT (fatty acids vs glucose vs amino acids, etc) are highly interesting.
Answer: Answer the above two questions together: The analysis of these problems has been added to the discussion of the manuscript. Briefly, KEGG analysis of RNA-seq (Fig. 4H) revealed that the difference of BCAA (valine, leucine and isoleucine) oxidation pathway and fatty acid metabolism pathway between neBAT and aBAT was significant, and highly enriched (Rich Factor). This suggests that neBAT enhances the catabolism of BCAA and fatty acids. Moreover, lipids and amino acids may be the substrate of BAT thermogenic, which is consistent with previous research results.
4) The data of present study is highly helpful for future studies; therefore, the protemic/trascriptome data should be uploaded in public database such as ProteomicsDB (https://www.proteomicsdb.org/) and ArrayExpress (https://www.ebi.ac.uk/arrayexpress/). To date, many of Journals requested authors to do so, because this will allow research communities to validate/re-analyze it, thereby increasing greatly the citation of the paper.
Answer: Thank you very much for your suggestion. We have prepared the data. When the journal intentionally accepts our manuscripts, we will upload it to the database in time.
5) As labeling of figure is too small and difficult to read, all figures including supplemental figures must be revised. Please enlarge labeling so that readers can see easily without zooming up.
Answer: We have updated the figures and put all the figures in PDF (attachment)
Minor points
6) Line 93: 1-3 days after pregnancy --> 1-3 days after “delivery” (if correct)
Answer: According to your comments, we have revised the text and checked all the manuscript.
7) Line 94: (after pregnancy) --> (after “delivery”) (if correct)
Answer: According to your comments, we have revised the text and checked all the manuscript.
8) Figure 1B: The authors concluded that volume of mitochondria increases in aBAT compared with neBAT (lines 221-222, 398-408). Please indicate mitochondria in Fig 1B by arrows and/or quantify the number of mitochondria per field.
Answer: We have marked the mitochondria of aBAT and neBAT with arrows in the figure 1B. It is obvious from the figure that the number of mitochondria gradually decreases with growth time course, but the volume gradually increases.
Reviewer 4 Report
The manuscript describes a deeply description and comparison of adult and neonatal brown adipose tissue of mice. The subject of the manuscript could fall within the scope of the Journal and interesting to publish, however there are several points that should be addressed.Against a lot of data produced, discussion and conclusion are very poor.
The English usage is often not correct and this makes difficult to understand the experiments and conclusion. The text should be improved by having input from a native English speaker. Authors should changes all the images at an higher magnification in order to allow the reader to see and understand them. Just some improvements are suggested here:
Abstract:
Should be more informative
Avoid the use of ‘etc’
Gene’s name should be in italic
Line 32: Please, re-wrote the sentence
Introduction:
Should be explain better the importance of this research rather than listen what the Authors performed
Lines 51-52: the sentence should be rewritten ‘is much better’ what does it mean?
Line 64: location marker?
Line 75:Surface morphology was examined by SEM not TEM
Material and Methods:
Lines 86-97:
Although the precise amounts and anatomical locations of BAT in mouse embryos have not been well document, There are a lot of different BAT depots in adult mice such as interscapular BAT and subscapular regions. Other small BAT depots are found deep in the dorsal neck between the scapula and the head (cervical BAT), around the aorta within the thoracic cavity (mediastinal BAT), and around the kidney (perirenal BAT) (Mo et al., JCI Insight. 2017 2(11): e93166).
From which site BAT was collected? Why data on embryonic group (how many animals?) was reported just as supplementary material?
Authors should include among the other results from neBAT and a BAT or completely delete them.
Goups of animals should be better described: 0 week old means 1-3 days after pregnancy; to which does 2 week old correspond? 4 weeks? 6 weeks?
Line 102: which Bodipy was used? No mention to Bodipy staining or data are reported in paper.
Lines 113-114: no sections should be done for SEM (nor postfixation is needed) at least the internal surface of cells must be observed. Is this the case? I guess no, from Fig. 1A.
Line 115: Which conductive material was used?
Line 118: how much RNA was extracted? How much used in cDNA synthesis? How good was the quality of RNA? Table 1 should indicate 5’-3’
Line 130: what protein concentration was used?
No paragraph on statistical analysis was described.
Results
The problem here is the quality of figures. Really, it is not possible to follow the results description which this images. Figure 1 lacks of scale bar.
Lines 212-218: how the Authors can state that the lipid droplets were smaller or uniform in size? Are the LDs measured? (I suggest Colitti et al., Eur J Histochem. 2018 Oct 25; 62(4): 2984). The same for the size of adipocytes.
Again for density of mitochondria: no mitochondria are distinguishable from Fig. 1B.Are mitochondria counted? Is there a statistical analysis about the count?
Line 247: the time of puberty should be stated in M&M when the sample collaction is described.
Discussion:
Discussion should be rewritten
Line 347: usually ‘internal and surface’ are not the correct term to indicate sections or SEM.
Line 403:this affermation should be reinforced by counts and statistical analysis
Line 412: ‘elevated expressed’ better ‘upregulated’.
Line 430-431: Maybe is the birth, just a physiological process, which trigger neonatal BAT in mice at week 0 in comparison to embryonic and adult, mice is more active because.
Lines 459-471: Sentence should be revised
Author Response
I attached it.

Round 2
Reviewer 2 Report
I had a look to the revised manuscript. I see the authors added additional analysis to better clarify their findings, while they did not consider minor points I raised. For example, all the reported western blots miss molecular weights; figure legend 1 indicates A) and B) but there is no A) and B) in the relative figure 1. Of course, it is obvious what they are referring to, but I think it is always better to be more accurate as possible. The same applies to the quantification I was asking; even it may be obvious, quantifications should be always reported. To be accurate is always an added value.
Reviewer 3 Report
The authors have addressed the most of my comments except one thing about presentation of the results. While labeling in the Figures has been improved, the characters have been still difficult to see because they are too small, and resolutions of the images is too low. The authors should improve the labeling (please enlarge more), and increase resolution (dpi) of the “original” image before transferring to PDF. I have no other comment.
Author Response
The authors have addressed the most of my comments except one thing about presentation of the results. While labeling in the Figures has been improved, the characters have been still difficult to see because they are too small, and resolutions of the images is too low. The authors should improve the labeling (please enlarge more), and increase resolution (dpi) of the “original” image before transferring to PDF. I have no other comment.
Answer: Thank you very much for your professional comments. We have improved the quality of the figures, enlarged them enough to be identified, and exported them to PDF files together with the figure legend. As for the interaction network (Figure 8 and Figure 9), it needs to be identified after amplification as the figure contains a lot of information. If appropriate, we can upload an independent PDF file of each interaction network figure as the extended material.
Reviewer 4 Report
This paper remains flawed, unfortunately, the revised paper is not good to be published in the present form. The replies are rough. Some results are not enclosed in the text, but only in Figures legend and are again unreadable. English should be improved at all.
Abstract:
ect or et cetera is the same: it should not be used in any part of a scientific paper!
Material and method
Line 112: ‘every 7 days is a week’ do the authors want to make fun about that? I asked about the number of animal. The answer was: “We collected the BAT from the interscapular of newborn mice. The number of mice in each group is 6, which has been explained in the figure legends.” Why the authors do not report the number in the text? Again, From which site BAT was collected is not reported in M&M.Results
Figure 1: LDs and mitochondria are marked with red arrows, but no captions (A, B) are reported on figures. Again scale bar in H&H is not visible. It is obvious from the figure that the lipid drops of neBAT are smaller and more uniform. It is not adequate in a scientific paper to write ‘is obvious’ when measurements should be done to verify the results. the activity of neBAT was not only higher than aBAT, but also higher than that ME-BAT, but the progenitor markers (Myf5 and EN1) in BAT were gradually decrease in the course of development from embryonic to adult, which provide a direction for our further research. So this result should be insert in the text. Nanogold: could the authors report the something more i.e. company… Moreover, nanogold is used to mark Ab instead of colloidal gold. Why did you use nanogold on SEM samples? The quality of RNA is not reported again. Please use the correct form to write primers sequences. Answer: Thank you very much for your comment. This may be a possibility, but the activity and quantity of BAT in human newborn fetuses is indeed stronger than in adults. This is a question worthy of study and comparison. We believe your comment is a very good thinking, and the real reason needs more further study. I am sorry but this answer is incomprehensible. Figs 3 to 8 are unreadable. Fig.2 Mice gene should be written in lowercase.Discussion
A lot of connections with Supplementary results are reported, but very few results are supported by literature. It seems that Authors performed a lot of work, but what they can infer is very poor. Please, consider and discuss better your results.
Author Response
I attached the file

Round 3
Reviewer 4 Report
The authors addressed part of my comments:
Figures should be improved again, resolution is too small. It would be better if every picture had an ident. (a, b, c….) Author mentioned a software to measured LDs and mitochondria, but not statistical analysis were done on these data. Graphs should be the same x-axis and y-axis. English could be improved (i.e. sections Abstract, animal care, Statistical analysis ….) Legends: genes should be written in italic and lower caseAuthor Response
Dear editor: We are glad to submit our revised manuscript titled “Comprehensive analysis of the characteristics and differences in adult and newborn brown adipose tissue (BAT): newborn BAT is a more active/dynamic BAT” (MS ID: cells-624859). We appreciate the time and efforts given by the editor and the reviewers. We have carefully revised our manuscript according to the editor and reviewer’s comments and all appropriate changes made in the revising manuscript. The revised content is clearly shown by the revised mode and the revised/renewed figures were located in each figure of the manuscript. The attachment contains three contents:Revised manuscript , answer to reviewer and Improved figures. We hope that we have addressed satisfactorily all the concerns by the reviewers in this revised version. If you have any further questions, please do not hesitate to contact us, and we will be very glad to make improvements. Sincerely yours Junyu Liu